# What Should a (Future) Deep Learning Theory Look Like? A Phenomenological Perspective

September 9, 2022

## Abstract

To advance deep learning methodologies in the next decade, a theoretical framework for reasoning about modern neural networks is needed. While efforts are increasing toward demystifying why deep learning is so effective, a comprehensive picture remains lacking, suggesting that a better theory is possible. We argue that a future deep learning theory should inherit three characteristics: a *hierarchically* structured network architecture, parameters *iteratively* optimized using stochastic gradient-based methods, and information from the data that evolves *compressively*. As an instantiation, we integrate these characteristics into a graphical model called *neurashed*. This model effectively explains some common empirical patterns in deep learning. In particular, neurashed enables insights into implicit regularization, information bottleneck, and local elasticity. Finally, we discuss how neurashed can guide the development of deep learning theories.

## 1 Introduction

Deep learning is recognized as a monumentally successful approach to many data-extensive applications in image recognition, natural language processing, and board game programs (Krizhevsky et al., 2017; LeCun et al., 2015; Silver et al., 2016). Despite extensive efforts (Jacot et al., 2018; Bartlett et al., 2017; Berner et al., 2021), however, our theoretical understanding of how this increasingly popular machinery works and why it is so effective remains incomplete. This is exemplified by the substantial vacuum between the highly sophisticated training paradigm of modern neural networks and the capabilities of existing theories. For instance, the optimal architectures for certain specific tasks in computer vision remain unclear (Tolstikhin et al., 2021).

To better fulfill the potential of deep learning methodologies in increasingly diverse domains, heuristics and computation are unlikely to be adequate—a comprehensive theoretical foundation for deep learning is needed. Ideally, this theory would demystify these black-box models, visualize the essential elements, and enable principled model design and training. A useful theory would, at a minimum, reduce unnecessary computational burden and human costs in present-day deep-learning research, even if it could not make all complex training details transparent.

Unfortunately, it is unclear how to develop a deep learning theory from first principles. Instead, in this paper we take a phenomenological approach that captures some important characteristics of deep learning. Roughly speaking, a phenomenological model provides an overall picture rather than focusing on details, and allows for useful intuition and guidelines so that a more complete theoretical foundation can be developed.

To address what characteristics of deep learning should be considered in a phenomenological model, we recall the three key components in deep learning: architecture, algorithm, and data (Zdeborová, 2020). The most pronounced characteristic of modern network architectures is their *hierarchical* composition of simple functions. Indeed, overwhelming evidence shows that multiple-layer architectures are superior to their shallow counterparts (Eldan & Shamir, 2016), reflecting the fact that high-level features are hierarchically represented through low-level features (Hinton, 2021; Bagrov et al., 2020). The optimization workhorse for

training neural networks is stochastic gradient descent or Adam (Kingma & Ba, 2015), which *iteratively* updates the network weights using noisy gradients evaluated from small batches of training samples. Overwhelming evidence shows that the solution trajectories of iterative optimization are crucial to generalization performance (Soudry et al., 2018). It is also known that the effectiveness of deep learning relies heavily on the structure of the data (Blum & Rivest, 1992; Goldt et al., 2020), which enables the *compression* of data information in the late stages of deep learning training (Tishby & Zaslavsky, 2015; Shwartz-Ziv & Tishby, 2017).

## 2 Neurashed

We introduce a simple, interpretable, white-box model that simultaneously possesses the *hierarchical, iterative*, and *compressive* characteristics to guide the development of a future deep learning theory. This model, called *neurashed*, is represented as a graph with nodes partitioned into different levels (Figure 1). The number of levels is the same as the number of layers of the neural network that neurashed imitates. Instead of corresponding with a single neuron in the neural network, an *l*-level node in neurashed represents a feature that the neural network can learn in its *l*-layer. For example, the nodes in the first/bottom level denote lowest-level features, whereas the nodes in the last/top level correspond to the class membership in the classification problem. To describe the dependence of high-level features on low-level features, neurashed includes edges between a node and its dependent nodes in the preceding level. This reflects the hierarchical nature of features in neural networks.

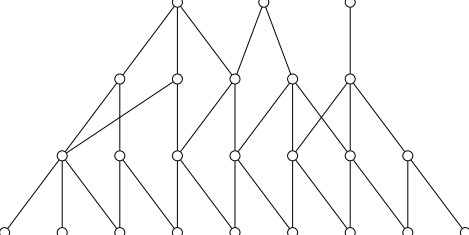

**Figure 1:** A neurashed model that imitates a four-layer neural network for a three-class classification problem. For instance, the feature represented by the leftmost node in the second level is formed by the features represented by the three leftmost nodes in the first level.

Given any input sample, a node in neurashed is in one of two states: firing or not firing. The unique last-level node that fires for an input corresponds with the label of the input. Whether a node in the first level fires or not is determined by the input. For a middle-level node, its state is determined by the firing pattern of its dependent nodes in the preceding levels. For example, let a node represent `cat` and its dependent nodes be `cat head` and `cat tail`. We activate `cat` when either or both of the two dependent nodes are firing. Alternatively, let a node represent `panda head` and consider its dependent nodes `dark circle`, `black ear`, and `white face`. The `panda head` node fires only if all three dependent nodes are firing.

We call the subgraph induced by the firing nodes the *feature pathway* of a given input. Samples from different classes have relatively distinctive feature pathways, commonly shared at lower levels but more distinct at higher levels. By contrast, feature pathways of same-class samples are identical or similar. An illustration is given in Figure 2.

To enable prediction, all nodes $F$ except for the last-level nodes are assigned a nonnegative value $\lambda_F$ as a measure of each node's ability to sense the corresponding feature. A large value of $\lambda_F$ means that when this node fires it can send out strong signals to connected nodes in the next level. Hence, $\lambda_F$ is the amplification factor of $F$. Moreover, let $\eta_{fF}$ denote the weight of a connected second-last-level node $f$ and last-level node $F$. Given an input, we define the score of each node, which is sent to its connected nodes on the next level: For any first-level node $F$, let score $S_F = \lambda_F$ if $F$ is firing and $S_F = 0$ otherwise; for any firing middle-level node $F$, we recursively define

$$S_F = \lambda_F \sum_{f \to F} S_f,$$

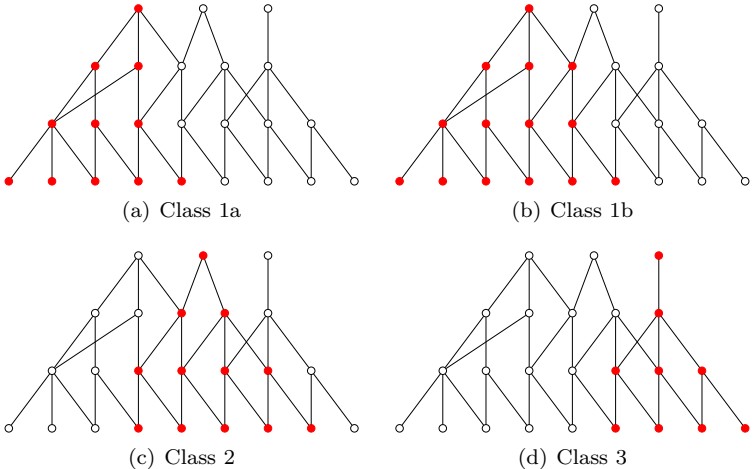

**Figure 2:** Feature pathways of the neurashed model in Figure 1. Firing nodes are marked in red. Class 1 includes two types of samples with slightly different feature pathways, which is a reflection of heterogeneity in real-life data (Feldman, 2020).

where the sum is over all dependent nodes $f$ of $F$ in the lower level. Likewise, let $S_F = 0$ for any non-firing middle-level node $F$. For the last-level nodes $F_1, \ldots, F_K$ corresponding to the $K$ classes, let

$$Z_j = \sum_{f \to F_j} \eta_{f F_j} S_f \tag{1}$$

be the logit for the $j$th class, where the sum is over all second-last-level dependent nodes $f$ of $F_j$. Finally, we predict the probability that this input is in the $j$th class as

$$p_j(x) = \frac{\exp(Z_j)}{\sum_{i=1}^{K} \exp(Z_i)}.$$

To mimic the iterative characteristic of neural network training, we must be able to update the amplification factors for neurashed during training. At initialization, because there is no predictive ability as such for neurashed, we set $\lambda_F$ and $\eta_{fF}$ to zero, other constants, or random numbers. In each backpropagation, a node is firing if it is in the *union* of the feature pathways of all training samples in the mini-batch for computing the gradient. We increase the amplification ability of any firing node. Specifically, if a node $F$ is firing in the backpropagation, we update its amplification factor $\lambda_F$ by letting

$$\lambda_F \leftarrow g^+(\lambda_F),$$

where $g^+$ is an increasing function satisfying $g^+(x) > x$ for all $x \geq 0$. The simplest choices include $g^+(x) = ax$ for $a > 1$ and $g^+(x) = x + c$ for $c > 0$. The strengthening of firing feature pathways is consistent with a recent analysis of simple hierarchical models (Poggio et al., 2020; Allen-Zhu & Li, 2020). By contrast, for any node $F$ that is *not* firing in the backpropagation, we decrease its amplification factor by setting

$$\lambda_F \leftarrow g^-(\lambda_F)$$

for an increasing function $g^-$ satisfying $0 \leq g^-(x) \leq x$; for example, $g^-(x) = bx$ for some $0 < b \leq 1$. This recognizes regularization techniques such as weight decay, batch normalization (Ioffe & Szegedy, 2015), layer normalization (Ba et al., 2016), and dropout (Srivastava et al., 2014) in deep-learning training, which effectively impose certain constraints on the weight parameters (Fang et al., 2021). Update rules $g^+, g^-$ generally vary with respect to nodes and iteration number. Likewise, we apply rule $g^+$ to $\eta_{fF}$ when the connected second-last-level node $f$ and last-level node $F$ both fire; otherwise, $g^-$ is applied.

The training dynamics above could improve neurashed's predictive ability. In particular, the update rules allow nodes appearing frequently in feature pathways to quickly grow their amplification factors. Consequently, for an input $x$ belonging to the $j$th class, the amplification factors of most nodes in its feature become relatively large during training, and the true-class logit $Z_j$ also becomes much larger than the other logits $Z_i$ for $i \neq j$. This shows that the probability of predicting the correct class $p_j(x) \to 1$ as the number of iterations tends to infinity.

The modeling strategy of neurashed is similar to a water*shed*, where tributaries meet to form a larger stream (hence "neura*shed*"). This modeling strategy gives neurashed the innate characteristics of a hierarchical structure and iterative optimization. As a caveat, we do not regard the feature representation of neurashed as fixed. Although the graph is fixed, the evolving amplification factors represent features in a dynamic manner. Note that neurashed is different from capsule networks (Sabour et al., 2017) and GLOM (Hinton, 2021) in that our model is meant to shed light on the black box of deep learning, not serve as a working system.

## 3   Insights into Puzzles

**Implicit regularization.** Conventional wisdom from statistical learning theory suggests that a model may not perform well on test data if its parameters outnumber the training samples; to avoid overfitting, explicit regularization is needed to constrain the search space of the unknown parameters (Friedman et al., 2001). In contrast to other machine learning approaches, modern neural networks—where the number of learnable parameters is often orders of magnitude larger than that of the training samples—enjoy surprisingly good generalization even *without* explicit regularization (Zhang et al., 2021a). From an optimization viewpoint, this shows that simple stochastic gradient-based optimization for training neural networks implicitly induces a form of regularization biased toward local minima of low "complexity" (Soudry et al., 2018; Bartlett et al., 2020). However, it remains unclear how implicit regularization occurs from a geometric perspective (Nagarajan & Kolter, 2019; Razin & Cohen, 2020; Zhou, 2021).

To gain geometric insights into implicit regularization using our conceptual model, recall that only firing features grow during neurashed training, whereas the remaining features become weaker during backpropagation. For simplicity, consider stochastic gradient descent with a mini-batch size of 1. Here, only *common* features shared by samples from different classes constantly fire in neurashed, whereas features peculiar to some samples or certain classes fire less frequently. As a consequence, these common features become stronger more quickly, whereas the other features grow less rapidly or even diminish.

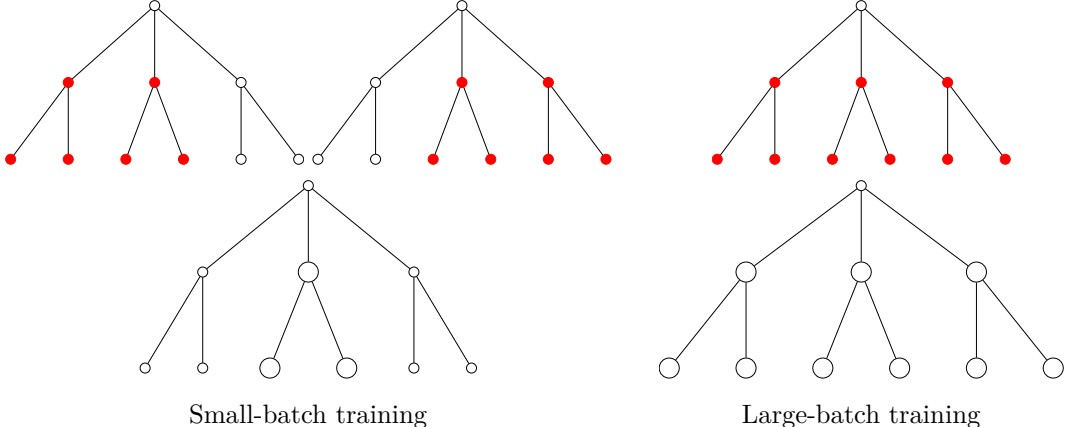

Small-batch training                    Large-batch training

**Figure 3:** Part of neurashed that corresponds to a single class. The two top plots in the left panel show two feature pathways, and the top plot in the right panel denotes the firing pattern when both feature pathways are included in the batch (the last-level node is firing but is not marked in red for simplicity). The two bottom plots represent the learned neurashed models, where larger nodes indicate larger amplification factors.

When gradient descent or large-batch stochastic gradient descent are used, many features fire in each update of neurashed, thereby increasing their amplification factors simultaneously. By contrast, a small-batch method constructs the feature pathways in a sparing way. Consequently, the feature pathways learned using small batches are *sparser*, suggesting a form of *compression*. This comparison is illustrated in Figure 3, which implies that different samples from the same class tend to exhibit vanishing variability in their high-level features during later training, and is consistent with the recently observed phenomenon of neural collapse (Papyan et al., 2020). Intuitively, this connection is indicative of neurashed's compressive nature.

Although neurashed's geometric characterization of implicit regularization is currently a hypothesis, much supporting evidence has been reported, empirically and theoretically. Empirical studies in Keskar et al. (2016); Smith et al. (2020) showed that neural networks trained by small-batch methods generalize better than when trained by large-batch methods. Moreover, Ilyas et al. (2019); Xiao et al. (2021) showed that neural networks tend to be more accurate on test data if these models leverage less information of the images. From a theoretical angle, HaoChen et al. (2020) related generalization performance to a solution's sparsity level when a simple nonlinear model is trained using stochastic gradient descent.

**Information bottleneck.** In Tishby & Zaslavsky (2015); Shwartz-Ziv & Tishby (2017), the information bottleneck theory of deep learning was introduced, based on the observation that neural networks undergo an initial fitting phase followed by a compression phase. In the initial phase, neural networks seek to both memorize the input data and fit the labels, as manifested by the increase in mutual information between a hidden level and both the input and labels. In the second phase, the networks compress all irrelevant information from the input, as demonstrated by the decrease in mutual information between the hidden level and input.

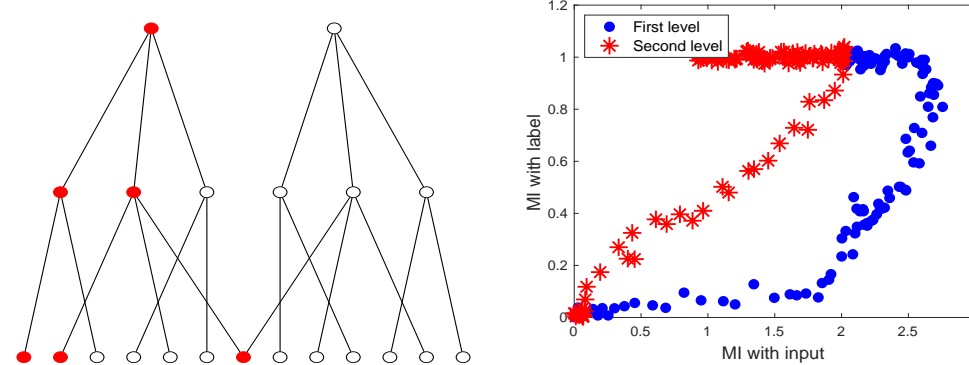

**Figure 4:** A neurashed model for a binary classification problem. All four firing patterns of Class 1 on the first level (from left to right): $(1, 2, 7), (2, 3, 7), (4, 5, 7), (5, 6, 7)$. In the second level, the first and third nodes fire if one or more dependent nodes fire, and the second (dominant) node fires if two or more dependent nodes fire. The left panel displays a feature pathway of Class 1. Class 2 has four feature pathways that are symmetric to those of Class 1. The right panel shows the information bottleneck phenomenon for this neurashed model. As with Shwartz-Ziv & Tishby (2017), noise is added in calculating the mutual information (MI) between the first/second level and the input (8 types)/labels (2 types). More details are given in the appendix.

Instead of explaining how this mysterious phenomenon emerges in deep learning, which is beyond our scope, we shed some light on information bottleneck by producing the same phenomenon using neurashed. As with implicit regularization, we observe that neurashed usually contains many redundant feature pathways when learning class labels. Initially, many nodes grow and thus encode more information regarding both the input and class labels. Subsequently, more frequently firing nodes become more dominant than less frequently firing ones. Because nodes compete to grow their amplification factors, dominant nodes tend to dwarf their weaker counterparts after a sufficient amount of training. Hence, neurashed starts to "forget" the information encoded by the weaker nodes, thereby sharing less mutual information with the input samples (see an illustration in Figure 4). The *compressive* characteristic of neurashed arises, loosely speaking, from the internal competition among nodes. This interpretation of the information bottleneck via neurashed is

reminiscent of the human brain, which has many neuron synapses during childhood that are pruned to leave fewer firing connections in adulthood (Feinberg, 1982).

**Local elasticity.** Last, we consider a recently observed phenomenon termed local elasticity (He & Su, 2020) in deep learning training, which asks how the update of neural networks via backpropagation at a base input changes the prediction at a test sample. Formally, for $K$-class classification, let $z_1(x, w), \ldots, z_K(x, w)$ be the logits prior to the softmax operation with input $x$ and network weights $w$. Writing $w^+$ for the updated weights using the base input $x$, we define

$$\mathrm{LE}(x, x') := \frac{\sqrt{\sum_{i=1}^{K}(z_i(x', w^+) - z_i(x', w))^2}}{\sqrt{\sum_{i=1}^{K}(z_i(x, w^+) - z_i(x, w))^2}}$$

as a measure of the impact of base $x$ on test $x'$. A large value of this measure indicates that the base has a significant impact on the test input. Through extensive experiments, He & Su (2020) demonstrated that well-trained neural networks are locally elastic in the sense that the value of this measure depends on the semantic similarity between two samples $x$ and $x'$. If they are similar—say, images of a cat and tiger—the impact is significant, and if they are dissimilar—say, images of a cat and turtle—the impact is low. Experimental results are shown in Figure 5. For comparison, local elasticity does not appear in linear classifiers because of the leverage effect. More recently, Chen et al. (2020); Deng et al. (2021); Zhang et al. (2021b) showed that local elasticity implies good generalization ability.

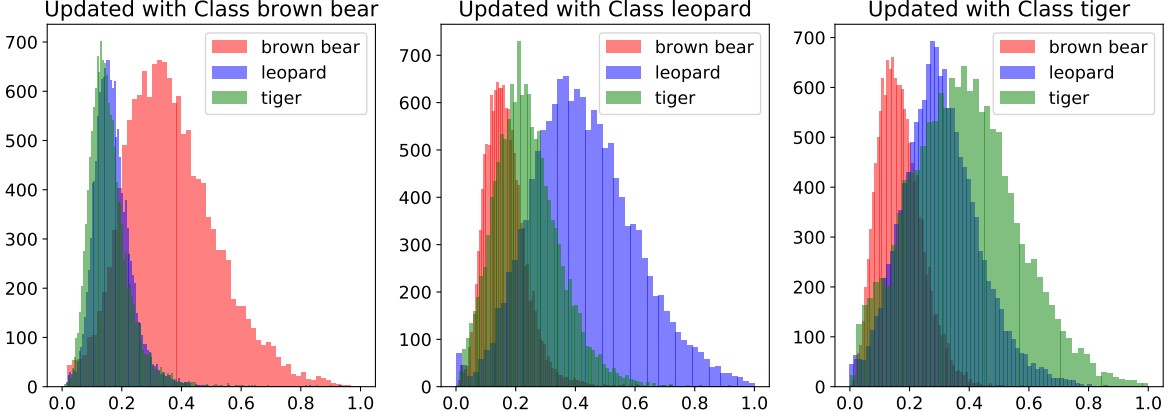

**Figure 5:** Histograms of $\mathrm{LE}(x, x')$ evaluated on the pre-trained VGG-19 network (Simonyan & Zisserman, 2014). For example, in the left panel the base input $x$ are images of brown bears. Each class contains 120 images sampled from ImageNet (Deng et al., 2009). Tiger and leopard are felines and similar.

We now show that neurashed exhibits the phenomenon of local elasticity, which yields insights into how local elasticity emerges in deep learning. To see this, note that similar training samples share more of their feature pathways. For example, the two types of samples in Class 1 in Figure 2 are presumably very similar and indeed have about the same feature pathways; Class 1 and Class 2 are more similar to each other than Class 1 and Class 3 in terms of feature pathways. Metaphorically speaking, applying backpropagation at an image of a leopard, the feature pathway for leopard strengthens as the associated amplification factors increase. While this update also strengthens the feature pathway for tiger, it does not impact the brown bear feature pathway as much, which presumably overlaps less with the leopard feature pathway. This update in turn leads to a more significant change in the logits equation 1 of an image of a tiger than those of a brown bear. Returning to Figure 2 for an illustration of this interpretation, the impact of updating at a sample in Class 1a is most significant on Class 1b, less significant on Class 2, and unnoticeable on Class 3.

## 4 Outlook

In addition to shedding new light on implicit regularization, information bottleneck, and local elasticity, neurashed is likely to facilitate insights into other common empirical patterns of deep learning. First, a byproduct of our interpretation of implicit regularization might evidence a subnetwork with comparable performance to the original, which could have implications on the lottery ticket hypothesis of neural networks (Frankle & Carbin, 2018). Second, while a significant fraction of classes in ImageNet (Deng et al., 2009) have fewer than 500 training samples, deep neural networks perform well on these classes in tests. Neurashed could offer a new perspective on these seemingly conflicting observations—many classes are basically the same (for example, ImageNet contains 120 dog-breed classes), so the effective sample size for learning the common features is much larger than the size of an individual class. Last, neurashed might help reveal the benefit of data augmentation techniques such as cropping. In the language of neurashed, `cat head` and `cat tail` each are sufficient to identify `cat`. If both concepts appear in the image, cropping reinforces the neurashed model by impelling it to learn these concepts separately. Nevertheless, these views are preliminary and require future consolidation.

While closely resembling neural networks in many aspects, neurashed is not merely intended to better explain some phenomena in deep learning. Instead, our main goal is to offer insights into the development of a comprehensive theoretical foundation for deep learning in future research. In particular, neurashed's efficacy in interpreting many puzzles in deep learning could imply that neural networks and neurashed evolve similarly during training. We therefore believe that a comprehensive deep learning theory is unlikely without incorporating the *hierarchical*, *iterative*, and *compressive* characteristics. That said, useful insights can be derived from analyzing models without these characteristics in some specific settings (Jacot et al., 2018; Chizat et al., 2019; Wu et al., 2018; Mei et al., 2018; Chizat & Bach, 2018; Belkin et al., 2019; Lee et al., 2019; Xu et al., 2019; Oymak & Soltanolkotabi, 2020; Chan et al., 2021).

Integrating the three characteristics in a principled manner might necessitate a novel mathematical framework for reasoning about the composition of nonlinear functions. Because it could take years before such mathematical tools become available, a practical approach for the present, given that such theoretical guidelines are urgently needed (E, 2021), is to better relate neurashed to neural networks and develop finer-grained models. For example, an important question is to determine the unit in neural networks that corresponds with a feature node in neurashed. Is it a filter in the case of convolutional neural networks? Another topic is the relationship between neuron activations in neural networks and feature pathways in neurashed. To generalize neurashed, edges could be fired instead of nodes. Another potential extension is to introduce stochasticity to rules $g^+$ and $g^-$ for updating amplification factors and rendering feature pathways random or adaptive to learned amplification factors. Owing to the flexibility of neurashed as a graphical model, such possible extensions are endless.

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

## Appendix

All eight feature pathways of the neurashed model in Figure 4. The left column and right column correspond to Class 1 and Class 2, respectively.

In the experimental setup of the right panel of Figure 4, all amplification factors at initialization are set to independent uniform random variables on $(0, 0.01)$. We use $g^-(\lambda_F) = 1.022^{-\frac{1}{4}}\lambda_F$ and $g^+(\lambda_F) = 1.022^{\frac{11}{4}}\lambda_F$ for all hidden nodes except for the 7th (from left to right) node, which uses $g^+(\lambda_F) = 1.022^{\frac{3}{4}}\lambda_F$. In the early phase of training, the firing pattern on the second level improves at distinguishing the two types of samples in Class 1, depending on whether the 1st or 3rd node fires. This also applies to Class 2. Hence, the mutual information between the second level and the input tends to $\log_2 4 = 2$. By contrast, in the late stages, the amplification factors of the 1st and 3rd nodes become negligible compared with that of the 2nd node, leading to indistinguishability between the two types in Class 1. As a consequence, the mutual information tends to $\log_2 2 = 1$. The discussion on the first level is similar and thus is omitted.

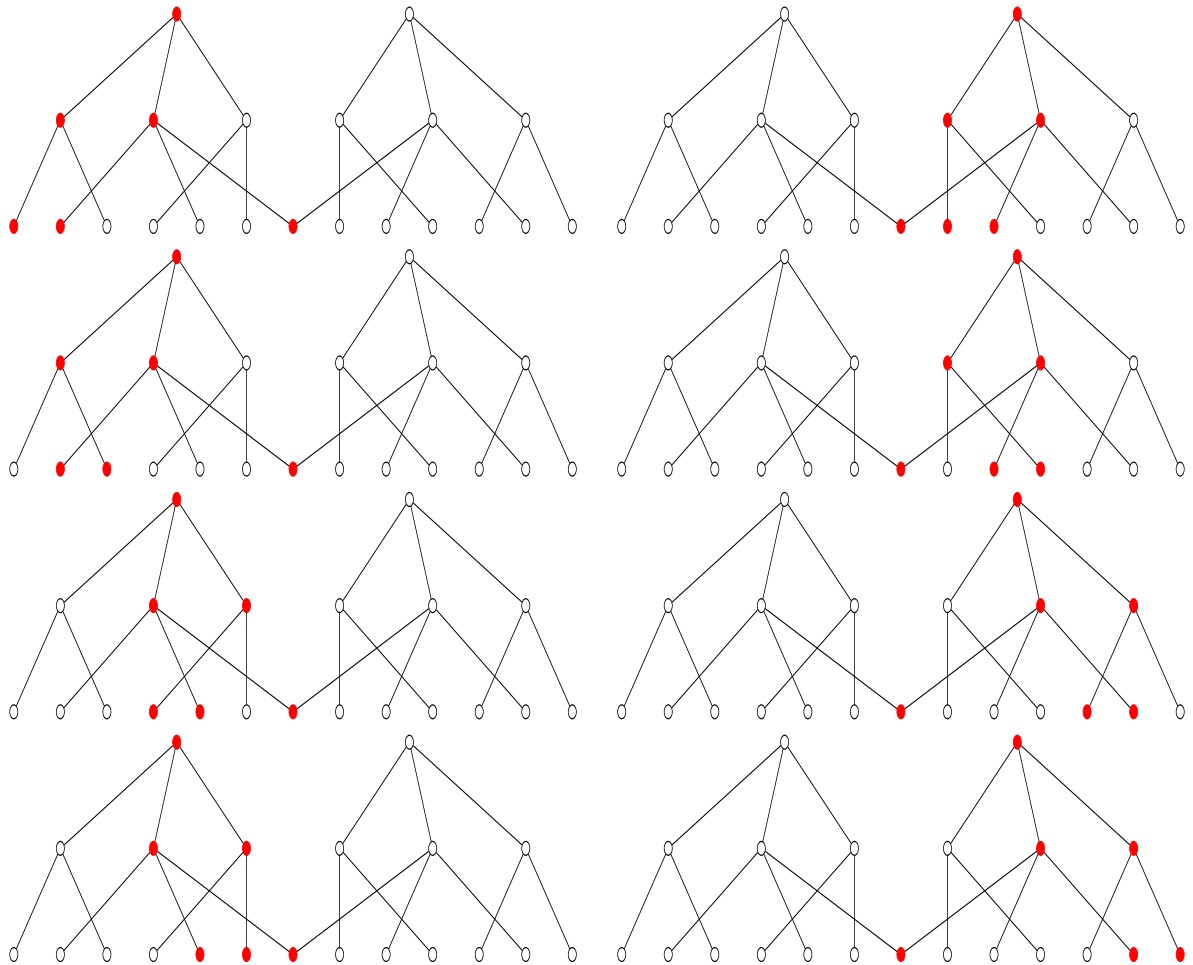

