# OpenReview forum: "What Should a (Future) Deep Learning Theory Look Like? A Phenomenological Perspective"
_TMLR — Rejected by TMLR_

### Review · Reviewer_Q35y · 2022-09-18

**Summary Of Contributions:**

The paper introduces a simplified model of a neural network model with units sending and receiving real scalar values and iteratively updating linear parameters. Authors argue that it is useful for studying implicit regularization, local elasticity and information compression found in real neural networks. Overall, I find this paper very confusing and not particularly insightful for "a future theory of deep learning".

**Broader Impact Concerns:**

No concerns

**Requested Changes:**

1. Present a very clear mathematical definition of a neurashaed.
2. Motivate the simpliciations, discuss relation of neurashed and neural networks. In which sense one speak for another?
3. Motivate the learning algorithm, discuss its properties in comparison to the standard backpropagation / gradient descent.
4. Make experiments fully reproducible.
5. Demonstrate that existing work is less efficient at studying the discussed phenomena than the presented work.

**Strengths And Weaknesses:**

# Strenghts:
* The idea of studying phenomena arising in neural networks using a simpler model makes sense to me.

# Weaknesses:

## Unclear connection between neural networks and neurasheds

From the sentence "The number of levels is the same as the number of layers of the neural network that neurashed imitates." it follows that a neural network induces a corresponding neurashed. If that's true, then I would appreciate such a mapping to be explicitly discussed. Especially in the sense of what insightful properties of the originan neural network are preserved by the neurashed and why.

## Why this model and this learning algorithm?

> In each backpropagation, a node is firing if it is in the union of the feature pathways of all training samples in the mini-batch for computing the gradient

* It's unclear what happens during the "forward pass" of the model, when does a node fire (if it happens just on a single object, i.e. batch size is 1)? It's impossible to reason about the model because the very first two equation involve this concept. In other words, the presentation of the method is confusing.
* Calling this algorithm "backpropagation" is very confusing becuase it has very little in common with computing gradients / derivatives or the authors should explain that.
* Why use this learning algorithm? It seems to be unnecessarily tied to the number of objects in the batch (see the quoted text), I strongly believe the learning dynamics for different batch sizes and sampling strategies will differ singnificantly and in undesired way.

##  Experiments are not presented clearly and difficult to reproduce

I failed at understanding how once can reproduce the presented experiments if they wanted to. I don't understand how were the neurasheds constructed for these experiments.

---

### Review · Reviewer_oTd5 · 2022-09-27

**Summary Of Contributions:**

The main contribution of the paper is the Neurashed framework. As argued, Neurashed, encoded three important factors of deep learning: hierarchical, iterative, and compressive characteristics. There are many frontier phenomena in deep learning explained by the Neurashed framework (mainly in section 3).

**Broader Impact Concerns:**

The reviewer does not foresee any ethical concerns of the current manuscript.

**Requested Changes:**

I would very much appreciate any discussions to the questions in the previous weakness section.

Overall, the paper is clearly written with very minor typos.

**Strengths And Weaknesses:**

A beauty, at least from my point of view, is the simplicity of the Neurashed framework. Moreover, it's structure is also intuitive to understand. Basically, Neurashed explores representation relation and models mutual influences between representation nodes.

Regarding weaknesses, I mainly have the following questions.
1. (Structural difference between Neurashed and a true neural networks). I am in fact curious about how a Neurashed can be constructed. In particular, how to determine nodes, how to choose the number of layers and the width. In the end, is it much different from a neural network it self? For example, we can evaluate the fire pattern of a well trained neural network. Does it correspond to any Neurashed construction?

2. (Future direction suggested by Neurashed) As introduced, Neurashed are proposed for future deep learning theory. From reading the paper, Neurashed is a good tool to explain (also not certain how to develop serious theory using Neurashed though) many interesting and controversial phenomena in practical deep learning. However, not much is discussed of Neurashed for future deep learning theory. Is it possible for Neurashed to suggest future deep learning directions? For example, how to train better? how to effectively choose network configurations, etc?

---

### Review · Reviewer_UnoE · 2022-10-03

**Summary Of Contributions:**

This paper proposes a new model for deep learning, termed the _neurashed_. This model shares some important features of neural networks, in particular their hierarchical structure, and can be trained via an iterative procedure that resembles stochastic gradient descent. The paper goes on to replicate three phenomena observed in deep learning in the neurashed model through thought experiments and empirical simulations: implicit regularization towards “simple” functions, compression of the learned representation over time, and _local elasticity_.


**Broader Impact Concerns:**

I do not have any broader impact concerns.

**Requested Changes:**

**Writing and clarity.**
A critical point that needs to be addressed before I can recommend acceptance is to clarify the precise formulation of the neurashed activation function and how it is trained. This is crucial for any paper which proposes a new learning algorithm, even if this algorithm is only intended as a phenomenological model of deep learning.

  - The clarity of the paper would be improved by including a short pseudocode algorithm summarizing how the updates are performed.

  - The relation to prior work on discrete learning rules, in particular Hebbian learning, should be discussed.

  - The rule for determining whether an intermediate node is turned on or off needs to be clarified.

  - Additional experiment details should be included in the supplementary materials.

**Theory.**
 - The paper does not currently demonstrate that the neurashed is more amenable to theoretical analysis than the neural network. Providing an example where this is the case, in a manner more rigorous than the thought experiments in Section 3, would significantly strengthen the claim that the neurashed can provide insight into deep neural networks.

 - Some analysis of the dynamics of the model showing that the update rule is a contraction or that it is guaranteed to reduce the objective would increase my confidence that the neurashed is a useful model of learning. Intuitively it seems like this should be the case depending on finding a suitable “learning rate”, but I would like to see this shown more rigorously.

 - At minimum, I would need to see at least a few examples of learning problems where the neurashed is able to learn a function that minimizes its training loss in order to be convinced that the neurashed is a reasonable model for learning.

**Experiments.**

 - Additional experiment details are needed to describe how Figure 4 is generated.

 - For both implicit regularization and local elasticity, the paper should demonstrate that these phenomena arise naturally from the learning dynamics of the neurashed to justify the claim that the neurashed model can provide insight into them.

**Strengths And Weaknesses:**


**Strengths**

 - There is scientific value in replicating phenomena of interest that appear in deep neural networks in other models of learning in order to better understand the necessary conditions for these phenomena.

 - The paper provides a number of visualizations that clarify structure and behaviour of the neurashed model, which I thought were particularly nicely presented.

 - The neurashed model captures a number of properties of neural networks that intiuitively seem to be important distinguishing factors from other machine learning systems, while allowing other features — in particular the optimization procedure — to vary. This is an appealing setup as it could also allow for further study into the role of gradient descent algorithms and the loss landscapes of neural network architectures separately from other properties of NNs, e.g. their hierarchical structure.
 - Figure 4 presents a compelling illustration that the neurashed model can replicate a relatively sophisticated property of neural network training, providing supporting evidence for its scientific utility.
 - The paper is for the most part clearly written (modulo some of my comments about the presentation of the neurashed model that follow), and effectively communicates its objective and conclusion.
 - The paper is ambitious in providing a new model of learning which, while sharing some important features with deep neural networks, follows significantly different learning dynamics. As a field, I think it is important that we consider a more diverse set of possible models of learning than solely neural networks trained with SGD-like optimizers, and this paper presents a preliminary but nonetheless intriguing step towards this diversification.


**Weaknesses**:

- Some details of the neurashed model remain unclear to me, and should be addressed in the main text of the paper:

    - The text describing the updating procedure for the neurashed is quite long, and the clarity of the paper would be improved by including a short pseudocode algorithm summarizing how the updates are performed.

    - The notion of ‘feature’ expressed by the nodes is unclear, as is the rule for determining whether a node is firing or not. On p2 there are two examples provided: one in which the activation function is an OR (the cat head/tail) and one in which it appears to be an AND (the panda head). The equation on the bottom of page 2, however, seems to be a continuous value which sums the incoming feature activations. This isn’t consistent with the logical activation functions discussed in the preceding paragraphs.
    - The neurashed update procedure features a number of design choices whose motivation is not discussed. This procedure resembles Hebbian learning, but I didn’t see any discussion of how this model relates to older models of learning which also use non-gradient update rules. The paper would benefit significantly from a discussion of related work and of the motivation behind the learning rule.

 - The neurashed model is proposed with the objective of providing a simplifying set of assumptions through which theorists can study deep learning. This is an admirable goal, however it isn’t clear to me that, for an equal level of expressiveness, a neurashed would be any more tractable  than a neural network. The examples provided in the paper consider sparse, narrow, and shallow architectures for which neural network behaviour would also be relatively tractable to analyze. However, scaling up this model to have similar expressive power as a large neural network would be computationally expensive, and I imagine would also make theoretical analysis difficult. A large part of the utility of a phenomenological model of deep learning lies in its simplicity, and so this strikes me as one of the major weaknesses of the neurashed.

 - An additional weakness of the neurashed model is that while it has been shown to replicate some phenomena of deep learning, it is not straightforward to deduce that these phenomena occur for the same reasons in the neurashed as they do in a deep neural network. The neurashed varies along a number of dimensions from a standard deep learning pipeline, and it’s possible that some of these sources of variation could serve as confounding factors. As a result, observing e.g. information compression curves in a neurashed does not necessarily immediately give us additional insight into the information bottleneck phenomenon in neural networks.

- While the neurashed model presents a new framework to think about hierarchical learning algorithms, the paper does not provide empirical or theoretical evidence that this model is capable of learning on even synthetic tasks. Details on the practical implementation of this model are also missing. Figure 4 seems to point towards the neurashed changing in some meaningful way over the course of training, but the paper would benefit significantly from an evaluation of the model’s performance on these tasks as well.

- The analysis of implicit regularization and local elasticity are relatively weak compared to that of the information bottleneck results. In the case fo implicit regularization, the mechanism by which the neurashed features are strengthened seems very different to what occurs in neural networks, and there is no empirical or theoretical analysis quantifying the degree of implicit regularization in the neurashed. The use of geometric in this context also doesn’t make sense to me, as in the setting of deep neural networks it generally refers to the structure of the loss landscape, which doesn’t have a straightforward analogue in a neurashed as it is not trained via gradient descent. Similarly, the replication of local elasticity is justified by an intuitive argument, but not quantified empirically.

---

### Decision · Action_Editors · 2022-11-18

**Recommendation:** Reject

**Comment:**

[see the claims and evidence above for a summary]

**Audience:**

People interested in getting better insights from neural networks.

**Claims And Evidence:**

The work proposes a new way of reasoning about "modern neural networks". To this end, the authors propose a new graphical model called "neurashed" which shares a lot of things with typical neural nets: hierarchical, optimized with SGD and "compressive" information flow. In general, the reviewers found the motivation for this work laudable, but the work itself ended bringing up more questions than answers.  One of the main drawbacks of the proposed model is that for a sufficiently expressive level of complexity, a neurashed wouldn't necessarily be more tractable to understand than a typical neural net. One of the reviewers mentions that "the paper does not provide empirical or theoretical evidence that this model is capable of learning on even synthetic tasks." Moreover, there are no details on the practical implementation of this model. While the work is motivated to improve theoretical understanding, not much is discussed in terms of improving deep learning theory.

As is, the work would need a lot of changes in order to pass the bar. All three reviews have plenty of suggestions of how to make this work better.